# Prevalences of Tobamovirus Contamination in Seed Lots of Tomato and Capsicum

**DOI:** 10.3390/v15040883

**Published:** 2023-03-30

**Authors:** David J. Dall, David A. Lovelock, Lindsay D. J. Penrose, Fiona E. Constable

**Affiliations:** 1Australian Government Department of Agriculture, Fisheries and Forestry, GPO Box 858, Canberra, ACT 2601, Australia; lindsay.penrose@aff.gov.au; 2Agriculture Victoria Research, AgriBio, 5 Ring Road, Bundoora, VIC 3083, Australia; david.lovelock@ecodev.vic.gov.au (D.A.L.); fiona.constable@ecodev.vic.gov.au (F.E.C.)

**Keywords:** tobamovirus, seed-borne pathogens, biosecurity

## Abstract

Seed lots of tomato and capsicum (*Solanum lycopersicon* and *Capsicum annuum*, respectively) are required to be free of quarantine pests before their entry to Australia is permitted. Testing of samples from 118 larger seed lots in the period 2019–2021 revealed that 31 (26.3%) carried one or more of four *Tobamovirus* species, including tomato mottle mosaic virus (ToMMV), which is a quarantine pest for Australia. Testing of samples from a further 659 smaller seed lots showed that 123 (18.7%) carried a total of five *Tobamovirus* species, including ToMMV and tomato brown rugose fruit virus (ToBRFV), which is also a quarantine pest for Australia. Estimated prevalence of contamination by tobamoviruses ranged from 0.388% to 0.004% in contaminated larger seed lots. Analyses of these data allow us to estimate probabilities of detection of contamination under different regulatory settings.

## 1. Introduction

Diseases caused by tobamoviruses (Genus *Tobamovirus*, Family Virgaviridae) can have significant effects on the growth and productivity of important food crops in the plant families Cucurbitaceae (e.g., melons and cucumbers) and Solanaceae (e.g., capsicums and tomatoes).

Many tobamoviruses are known to be seed transmissible, providing a direct route for the establishment of seedling/plant foci of infection in production settings. In the case of ToBRFV, transmission via seeds from experimentally infected tomato plants has been reported at a rate of 2.8%, and 1.8% for cotyledons and third true leaves of seedlings, respectively [1]; transmission has also been reported via seeds originating from field-infected tomato fruit, albeit at a lower rate of 0.08% [2]. From such foci, the viruses can rapidly spread by direct mechanical transmission between plants, and via other contact activities with humans, equipment and pollinating insects [3]. Once infection is established, there are no effective remedial treatments, and plant/crop destruction is a common practice for minimizing further dissemination. Tobamoviruses are also able to persist in viable form for lengthy periods in soil and aqueous environments, making them additionally difficult to manage and/or eradicate once present at a site [4].

A key element of biosecurity protection against quarantine-status tobamoviruses is to minimise, to an appropriate level, the likelihood of their entry through the seeds-for-planting pathway. Many jurisdictions adopt regulatory measures for this purpose, commonly involving testing of seed samples for non-detectability of quarantine *Tobamovirus* species; polymerase chain reaction (PCR) technologies are commonly used for detection, and in some jurisdictions serological (e.g., ELISA) and/or bioassay protocols are also accepted.

Differences also exist between jurisdictions with respect to the intensity of seed sampling required; in some instances, agencies advocate, and jurisdictions accept, testing of samples of 3000 seeds from seed lots proposed for importation [5,6]. Australia mandates PCR testing of the lesser of either 20% or 20,000 seeds from tomato and capsicum seed lots proposed for importation, and requires non-detectability of quarantine tobamoviruses, such as tomato brown rugose fruit virus (ToBRFV) and tomato mottle mosaic virus (ToMMV). Under circumstances in which generic PCR primers and sequencing of resultant amplicons are used, the identification of other non-quarantine tobamoviruses present in seed lots is also possible.

Theoretical estimations indicate that testing of 20,000 seeds gives a probability of detection of 0.99 for the presence of virus contamination at a level of 0.023%, and a higher probability at rates above this level. However, in the absence of empirical information about ‘real-world’ levels of contamination of seed lots, it is difficult to assess the appropriateness of this (or any other) regulatory requirement.

Australia was among the first countries in the world to introduce emergency measures for the presence of ToBRFV and ToMMV in seeds of tomato and capsicum, commencing in March and November 2019, respectively. This study therefore presents benchmark data and analyses of the frequencies and prevalence of *Tobamovirus* contamination in seed lots of tomato and capsicum seed-for-planting, providing an overview of the virological status of the global seed production industries, profiles of contamination prevalence, and a basis from which to consider the effectiveness of different regulatory settings and their consequences for post-entry production.

This study further compares the contamination rates of tobamoviruses observed during the period 2019–2021, with rates of *Pospiviroid* contamination assessed in seed lots of tomato and capsicum in the period 2009–2016 [7,8]. As for tobamoviruses, pospiviroids can be vertically transmitted from contaminated seeds to germinating seedlings and resultant plants, and five species of pospiviroid are recognised as quarantine pests for Australia. The previously reported [7,8] empirically derived data for *Pospiviroid* contamination rates in tomato and capsicum seed lots represent, to our knowledge, the only existing dataset comparable to that for tobamoviruses presented in this report.

## 2. Materials and Methods

Data reported here for samples from ‘smaller’ seed lots are based on analyses of 659 seed lots (431 of tomato, 228 of capsicum) tested over the three-year period 2019–2021. Seed samples ranging from approximately 400 seeds to 8000 seeds were analysed for each smaller seed lot; the originating seed lots thus ranged from total sizes of about 2000 seeds up to 40,000 seeds, and on that basis were considered to be seeds largely intended for breeding, seed increase and other field trialling purposes.

Data reported for samples from ‘larger’ seed lots are based on analyses of a total of 118 seed lots tested over the same three-year period. Samples of at least 10,000 seeds were analysed for each larger seed lot; the originating seed lots were thus of a minimum size of 50,000 seeds, and on that basis were considered to be seed lots intended for commercial evaluation and production purposes. Samples from seventy-one large seed lots of tomato, comprising a total of 1,281,600 seeds (mean of 18,051 seeds per sample) and 47 larger seed lots of capsicum (760,800 seeds total; mean of 16,187 seeds) were tested.

Laboratory protocols were broadly as previously reported [9]. Briefly, sub-samples of 400 seeds were individually processed (using up to 50 sub-samples per seed lot), with extracted nucleic acids assayed by conventional reverse-transcription PCR (RT-PCR) using *Tobamovirus*-directed primers [10]. Direct Sanger sequencing of resultant amplicons [9] allowed determination of the specific identities of all detected tobamoviruses. All seed extracts were also assessed for the presence of pospiviroids using protocols previously described [7].

The virus prevalence (fraction of contaminated seeds) within each contaminated larger seed lot was estimated using a Markov Chain Monte Carlo method, assuming a Bayesian hierarchical model with a compound Bernoulli-Hypergeometric distribution. The hypergeometric distribution models the number of infected seeds sampled within a lot, while the Bernoulli distribution models the probability of a sample being detected as infected [7].

## 3. Results and Discussion

### 3.1. Incidences of Contamination in Smaller Seed Lots

The incidences and identities of tobamoviruses detected in samples from smaller seed lots are shown in Table 1. Data are binned according to the numbers of seed tested per sample, representing originating seed lots ranging from very small sizes (up to 2000 seeds in total) through two groupings of seed lots of 4000–10,000 seeds and 12,000–40,000 seeds. As shown, the most commonly detected viruses were ToBRFV (32 detections) and ToMMV (39 detections), both of which are quarantine pests for Australia, as well as for many other countries [11,12]. Tomato mosaic virus (ToMV) and the ubiquitous pepper mild mottle virus (PMMoV)[13] were each detected on 25 occasions.

Most prominent among these data are the observed high levels of incidence of ToBRFV and ToMMV in samples from very small lots of tomato seeds, significantly contributing to the observed overall contamination incidence of 25.3% in this sub-group. Noting that seed lots of this size are generally considered to represent propagules of plant breeding lines, there appears to be the possibility of significant levels of contamination of commercial germplasm with these viruses—an observation consistent with one previously made by Fowkes et al. [14].

Testing of the same seed samples for the presence of pospiviroids detected the presence only of potato spindle tuber viroid, which was found on 15 occasions in seed lots of capsicums.

### 3.2. Incidences and Prevalences of Contamination in Larger Seed Lots

The incidences of tobamoviruses detected in samples from larger seed lots are shown in Table 2. As shown, the most commonly detected virus was ToMV, with an overall combined detection rate in tomato and capsicum seed lots of 12.7%; the highest individual incidence was of PMMoV in capsicum seed lots, with a detection rate of 21.3%. Of particular significance was the detection on four occasions of ToMMV, which was recorded from seed lots of both tomato and capsicum.

All larger seed lots were also tested for the presence of pospiviroids; only one detection was made, being an instance of tomato apical stunt viroid in a capsicum seed lot.

Figure 1 provides a cumulative distribution curve of *Tobamovirus* prevalences for the combined data for larger tomato and capsicum seed lots (see Appendix A), together with the corresponding curve for previously estimated *Pospiviroid* prevalences in the same hosts [8]. The *Tobamovirus* curve comprises 29 data points; prevalence estimates could not be derived for one tomato seed lot from which all 50 of the 50 sub-samples returned a positive result for ToMMV, and a capsicum seed lot for which all 30 sub-samples were positive for PMMoV.

The comparison shows that the two cumulative distribution curves possess similar forms, indicating the presence, in both *Tobamovirus*- and *Pospiviroid*-contaminated seed lots, of a relatively small number of heavily contaminated lots and a larger number of more sparsely contaminated lots. On a quantitative basis, the levels of contamination were also broadly comparable for the two pathogen groups, ranging from 0.388% to 0.004% for tobamoviruses (the latter being the lowest possible estimation for the analytical procedure used), and from 0.476% to 0.004% for pospiviroids. The median contamination prevalences were 0.014% and 0.024% for tobamoviruses and pospiviroids, respectively, (the former with inclusion of the two non-estimable contaminated samples noted above and recognising that population sizes differ between the two groups).

Statistical calculations using the binomial distribution show that sample sizes of 3000 and 20,000 seeds allow detection of virus prevalences as low as 0.100% and 0.015%, respectively, with 95% confidence (see Figure 2). The calculations further indicate that the same sample sizes allow detection of virus prevalences as low as 0.150% and 0.023%, respectively, with 99% confidence (see Figure 3).

Figure 2 provides an overlay of the calculated 95% confidence level thresholds for those two sample sizes on the observationally derived cumulative distribution curves. For each threshold line, the proportion of the contamination curve lying to its right can be detected with 95% confidence. As shown, for a 20,000 seed sample the most heavily contaminated 50–60% of the contaminated seed lot populations are predicted to be detectable; in contrast, use of a 3000 seed sample is predicted to detect only some 11–16% of the same populations.

Noting the relatively high risk posed by viruses and viroids associated with seeds for planting, a detection confidence level of 99% is considered to be more appropriate than the lower level of 95%. Figure 3 provides an overlay of the calculated 99% confidence level thresholds for the same two sample sizes on the same cumulative distribution curves. As shown, for a 20,000 seed sample the most heavily contaminated 35–50% of the contaminated seed lot population is predicted to be detectable, while use of a 3000 seed sample is predicted to detect only some 10% of the same population.

In overview, the difference in detection capacity for the two sample sizes therefore spans approximately 25% to 40% of seed lots distributed between the 10th and 50th percentiles, as ranked by level of contamination. Given the rapid diminution of detection confidence for the 3000 seed sample size with reducing contamination rate [8], there is a high likelihood that a substantial proportion of contaminated lots—particularly in the more lightly contaminated section of the 10th–50th percentile span—would not be detected using such a sample.

As noted previously, Australia uses a 20,000 seed test for large seed lots; recognising that global seed production commonly involves multiple processes across different jurisdictions, this testing requirement is applied irrespective of the nominal country of consignment origin. To date, Australia has also not recorded an on-shore incursion of either ToBRFV or ToMMV. This is in contrast to reported incursion events in a number of other countries which adopt a standard based around 3000 seed sample testing [11,12]. These divergences in policy settings and observed outcomes thus prospectively offer an opportunity for assessment of the situation in the context of a ‘natural experiment’, sensu McKenna and Morrison [15], who broadly define natural experiments as ‘observational studies which can … assess the outcomes and impacts of policy interventions.’

Prima facie, it is possible to interpret the contrasting national experiences with ToBRFV and ToMMV noted above as consequences of differences in seed testing policies/stringencies, with this hypothesis further supported by the data and analysis presented in this study. Accepting this interpretation at face value, two potentially assessable corollaries emerge:For a given pest species, jurisdictions that impose more rigorous at-border testing regimes for seed contamination should record fewer incidences of post-border pest incursions than those using less rigorous testing regimes; andFor a given jurisdiction, pests that are subject to more rigorous at-border testing regimes should display lower incidences of post-border incursion than those subject to less rigorous regimes.

In practice, it is apparent that rigorously assessing such corollaries for causality is difficult, if not essentially impossible, consistent with the acknowledgement by McKenna and Morrison [15] that “natural experiments will never *unequivocally* (sic) determine causation because the researcher cannot exert control over the situation.” In the current setting, we acknowledge that other pertinent factors may be involved in, or perhaps even largely explain observed incursion incidences and frequencies. Thus, for example, with respect to the first corollary, normalised national patterns of cross-border seed movement on a volume-per-year basis, which could be considered to be relevant, are not taken into account. Likewise, for the second, a range of pest- and host-specific factors, of potentially locally variable importance, may affect critical processes of pest establishment and spread which precede an incursion event.

Nonetheless, as additional data accumulate across time, jurisdictions and pest identities, it is likely that it will be possible to draw ever more confident inferences about the roles of different seed testing regimes and policies on patterns of global transmission of seed-borne pathogens.

## Figures and Tables

**Figure 1 viruses-15-00883-f001:**
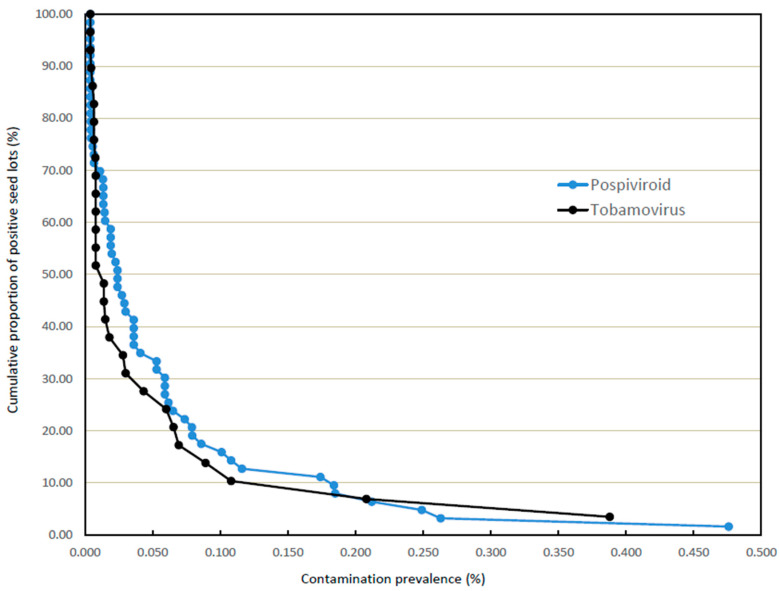
Cumulative distribution curves of tobamovirus and pospiviroid prevalences in contaminated tomato and capsicum seed lots. Curves display the cumulative proportions by percentage (*y*-axis) of contaminated seed lots with pathogen prevalence greater than, or equal to, to the fractional rate (%) of prevalence identified on the x-axis.

**Figure 2 viruses-15-00883-f002:**
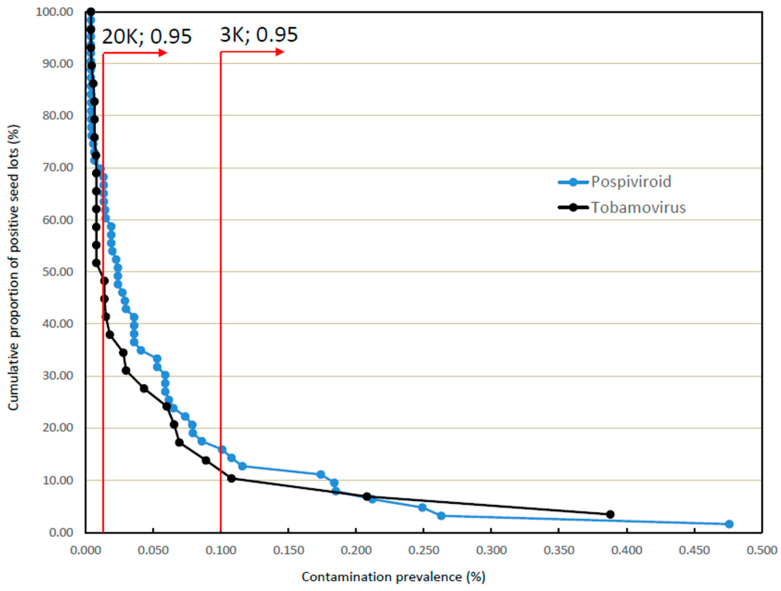
Comparative detectability of contaminated consignments with 95% confidence using test sample sizes of 20,000 and 3000 seeds.

**Figure 3 viruses-15-00883-f003:**
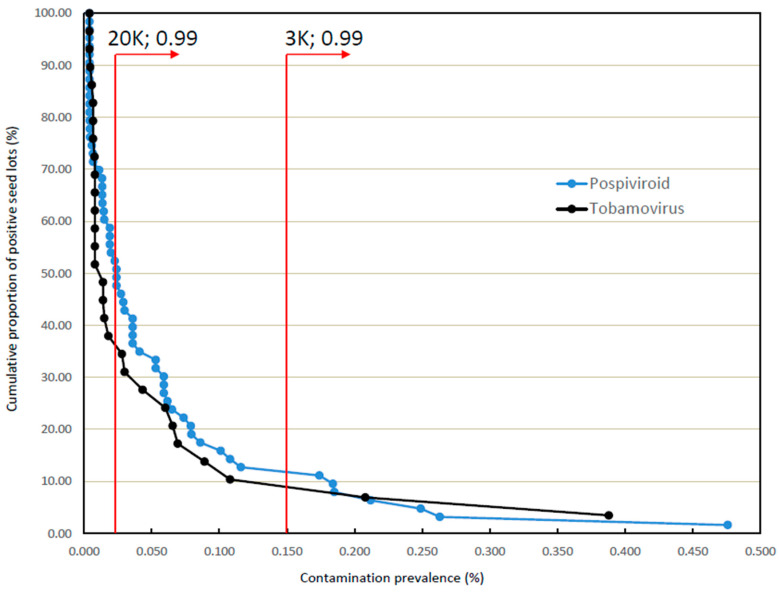
Comparative detectability of contaminated consignments with 99% confidence using test samples sizes of 20,000 and 3000 seeds.

**Table 1 viruses-15-00883-t001:** Identities and incidences of tobamoviruses in samples from smaller seed lots of tomato and capsicum, 2019–2021.

Virus Identity ^a.^	Tomato			Capsicum			Combined		
	≤400 ^b^	800–2000 ^b^	2400–8000 ^b^	≤400 ^a^	800–2000	2400–8000	≤400 ^a^	800–2000	2400–8000
PMMoV	1	2	2	8	5	7	9	7	9
TMV	0	0	1	0	1	0	0	1	1
ToMV	6	10	4	0	2	3	6	12	7
ToBRFV ^c^	23	4	3	0	0	2	23	4	5
ToMMV ^c^	25	4	2	6	0	2	31	4	4
Total detections	55	20	12	14	8	14	69	28	26
Total tested	217	140	74	74	91	63	291	231	137
% contaminated	25.3	14.3	16.2	18.9	8.8	22.0	23.7	12.1	19.0

^a^ PMMoV, Pepper mild mottle virus; TMV, Tobacco mosaic virus; ToMV, Tomato mosaic virus; ToBRFV, Tomato brown rugose fruit virus; ToMMV, Tomato mottle mosaic virus. ^b^ Numbers of seeds tested per seed lot; see text for further details. ^c^ Quarantine pest for Australia.

**Table 2 viruses-15-00883-t002:** Identities and incidences of tobamoviruses in samples from larger seed lots of tomato and capsicum, 2019–2021.

	Tomato		Capsicum		Combined	
Virus Identity ^a^	Total Detections	% Contaminated	Total Detections	% Contaminated	Total Detections	% Contaminated
PMMoV	1	1.4	10	21.3	11	9.3
TMV	0	–	1	2.1	1	0.8
ToMV	7	9.9	8	17.0	15	12.7
ToMMV ^b^	3	4.2	1	2.1	4	3.4
Total detections	11	15.5	20	42.6	31	26.3
Total tested	71		47		118	

^a^ see Table 1 for virus identities. ^b^ Quarantine pest for Australia.

## Data Availability

Not applicable.

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
