# Peer review of "Prevalences of Tobamovirus Contamination in Seed Lots of Tomato and Capsicum"

_viruses, 2023, doi:10.3390/v15040883_

Round 1

Reviewer 1 Report

The manuscript (Communication) with the title “Prevalences of Tobamovirus Contamination in Seed Lots of Tomato and Capsicum” by Dall et al.  presents data on the analysis of seed samples in an effort to generate relevant information for regulatory settings.

This type of notes might provide the basis for the decision-making process in quarantine regulation.

Some suggestions. 

In the introduction, they suddenly mention the comparison with samples tested for pospiviroids. For many readers, it might not be clear why. The authors might want to explain a bit more about this comparison. Any biological resemblance? Or that data was just available? Etc.

Discussion.

The two corollaries are clear. 

However, the discussion does not include the importance (?) of the origin of the seed (source).

For example. Seeds can come from Source A and Source B (or country 1 and country 2).If source A has a history of previous detection (reports) of a specific pathogen (company, geographic area, etc.), and Source B has never reported that pathogen, will these antecedents justify a larger sample to be analyzed in case A? Countries do have restrictions based on the presence of the specific pathogen in the geographic area, but is this something that can be address on rigorous method as described in the note?

It will be interesting, and might improve the note, if the authors consider this in the discussion. The last paragraph mentions “geographic space” but I am not sure if they are referring to the example above.

On the technical side. It will be interesting to mention that all protocols for pathogen detection do have similar sensitivities. Perhaps, this point has been addressed in previous publications of the group, but it will be important to mention that here. A more rigorous method might not only include larger samples but also more sensitive protocols.

Author Response

Please see the attachment.  Please note that line numbers quoted in the text in each attachment correspond to those of the 'Manuscript for Revisions' returned to us; for ease of reference our manuscript revisions are provided as tracked changes. 

Reviewer 2 Report

1. Line 12 describing the virus as a quarantine pest is incorrect, and there are several misdescriptions below. Maybe you're trying to describe a virus carried by a pest.

2. Line 80 what you mean with “Sequencing of resultant amplicons allowed determination of the specific identities of all detected tobamoviruses” ? How are so many samples sequenced?

3. Was disinfection done when testing the seeds? Is it possible to distinguish whether the virus detected is in the seed coat or inside the embryo? How to ensure the reliability of the results?

4. For the detection of these viruses, was there any consideration of vertical transmission also known as transmission from seed to seedling?

Reviewer 3 Report

the data of contamination rates of tobamoviruses observed only in 2019-2021,authors should add more data, for exzample 2017-2022 or 2018-2022....

A minor change is required in the PDF.

Reviewer 4 Report

This paper is a large-scale study of tobamovirus contamination rates in Solanaceae crop (tomato and capsicum) seed lots. The results presented in this study provide important insights into the quarantine of tomato and capsicum seeds during import and export. The methods follow the methodology previously reported by the authors, and the paper is well-written. I think it is worth publishing in the Viruses.

Author Response

The authors thank Reviewer 4; no further response required.

Round 2

Reviewer 2 Report

The frame design of the article is too simple, just to detect the viral type.

The experimental design needs to go a little deeper for the study to be meaningful.

There is no novelty and originality.

Reviewer 3 Report

The manuscript is well written, and it is suitable for publication. There are some suggestions and questions in the attached PDF file of the manuscript.
